

# The complete chloroplast genome sequence of *Gentiana lawrencei* var. *farreri* (Gentianaceae) and comparative analysis with its congeneric species

Peng-Cheng Fu[1], Yan-Zhao Zhang[1], Hui-Min Geng[1] and Shi-Long Chen[2]

[1] College of Life Science, Luoyang Normal University, Luoyang, China
[2] Key Laboratory of Adaptation and Evolution of Plateau Biota, Northwest Institute of Plateau Biology, Chinese Academy of Sciences, Xining, China

## ABSTRACT

**Background**. The chloroplast (cp) genome is useful in plant systematics, genetic diversity analysis, molecular identification and divergence dating. The genus *Gentiana* contains 362 species, but there are only two valuable complete cp genomes. The purpose of this study is to report the characterization of complete cp genome of *G. lawrencei* var. *farreri*, which is endemic to the Qinghai-Tibetan Plateau (QTP).

**Methods**. Using high throughput sequencing technology, we got the complete nucleotide sequence of the *G. lawrencei* var. *farreri* cp genome. The comparison analysis including genome difference and gene divergence was performed with its congeneric species *G. straminea*. The simple sequence repeats (SSRs) and phylogenetics were studied as well.

**Results**. The cp genome of *G. lawrencei* var. *farreri* is a circular molecule of 138,750 bp, containing a pair of 24,653 bp inverted repeats which are separated by small and large single-copy regions of 11,365 and 78,082 bp, respectively. The cp genome contains 130 known genes, including 85 protein coding genes (PCGs), eight ribosomal RNA genes and 37 tRNA genes. Comparative analyses indicated that *G. lawrencei* var. *farreri* is 10,241 bp shorter than its congeneric species *G. straminea*. Four large gaps were detected that are responsible for 85% of the total sequence loss. Further detailed analyses revealed that 10 PCGs were included in the four gaps that encode nine NADH dehydrogenase subunits. The cp gene content, order and orientation are similar to those of its congeneric species, but with some variation among the PCGs. Three genes, *ndhB*, *ndhF* and *clpP*, have high nonsynonymous to synonymous values. There are 34 SSRs in the *G. lawrencei* var. *farreri* cp genome, of which 25 are mononucleotide repeats: no dinucleotide repeats were detected. Comparison with the *G. straminea* cp genome indicated that five SSRs have length polymorphisms and 23 SSRs are species-specific. The phylogenetic analysis of 48 PCGs from 12 Gentianales taxa cp genomes clearly identified three clades, which indicated the potential of cp genomes in phylogenetics.

**Discussion**. The "missing" sequence of *G. lawrencei* var. *farreri* mainly consistent of *ndh* genes which could be dispensable under chilling-stressed conditions in the QTP. The complete cp genome sequence of *G. lawrencei* var. *farreri* provides intragenic information that will contribute to genetic and phylogenetic research in the Gentianaceae.

Corresponding author
Shi-Long Chen, slchen@nwipb.cas.cn

## INTRODUCTION

The chloroplast (cp) is the photosynthetic organelle that provides essential energy for plants, and is hypothesized to have arisen from ancient endosymbiotic cyanobacteria (*Neuhaus & Emes, 2000*). In angiosperms, most cp genomes are circular DNA molecules, containing one large single-copy region (LSC), one small single-copy region (SSC) and a pair of inverted repeats (IRs) (*Palmer, 1985*; *Jansen et al., 2005*). The sizes of cp genomes in most angiosperms range from 120 kb to 160 kb caused by expansion of the IR regions and evolutionary contractions (*Palmer, 1985*; *Wang et al., 2008*).

Recently, the number of completely sequenced cp genomes from higher plants has increased significantly. The cp genome is useful in plant systematics research because of its maternal inheritance, haploid nature and highly conserved structures. It is widely used in the study of genetic diversity, molecular identification, phylogenetic classification and divergence dating (*Shaw et al., 2007*; *Nikiforova et al., 2013*; *Carbonell-Caballero et al., 2015*; *Williams et al., 2016*). The comparative analysis of cp genomes reveal insights into the cp genome evolution such as sequence inversion (*Cho et al., 2015*), gene loss (*Wakasugi et al., 1994*; *Millen et al., 2001*) and variation in borders of LSC, SSC and IR regions (*Ni et al., 2016*).

The family Gentianaceae has approximately 700 species (*He, 1988*) and is the third largest family of the Gentianales order in the Asterids clade. However, only one complete chloroplast genomes has been reported in this family so far (*Ni et al., 2016*). *Gentiana* is the largest genus in the Gentianaceae, containing 15 sections and about 362 species (*Ho & Liu, 2001*). *Gentiana* plants have been widely used as traditional Chinese and Tibetan medicines (*Ho & Liu, 2001*) and are edificators in the Qinghai-Tibetan Plateau (QTP) alpine meadow. Although some studies have been carried out on the phylogenetics of *Gentiana*, they have all been based on one or several gene fragments (*Yuan & Küpfer, 1997*; *Yuan, Kupfer & Doyle, 1996*; *Zhang et al., 2009*). Together with their complicated evolutionary history (*Yuan & Küpfer, 1997*), the phylogenetic relationships of *Gentiana*, especially intrasectional classification, remain controversial (*Ho & Liu, 2001*; *Favre et al., 2010*). At present, there are only two complete cp genomes have been sequenced in the *Gentiana*: *G. straminea* and *G. crassicaulis*, which both belong to the same section, *Cruciata* Gaudin, and only *G. straminea* was reported (*Ni et al., 2016*). Therefore, it is necessary to develop genomic resources for *Gentiana* to provide valuable information to study their phylogenetic relationships and the evolutionary history of the genus.

*Gentiana lawrencei* var. *farreri* T. N. Ho is endemic to the QTP and belongs to sect. *Kudoa* (Masamune) Satake & Toyokuni ex Toyokuni. It has very beautiful flowers and has been used in traditional Chinese and Tibetan medicine (*Yang et al., 2012*). Here, we report the cp genome sequence of *G. lawrencei* var. *farreri* and present a comparative analysis with its congeneric species *G. straminea* The genome structure, insertions and deletions, repeat sequences and phylogenetics of Gentianaceae were analyzed. This study provided large amounts of sequence information for phylogenetic and evolutionary studies of *Gentiana* and the Gentianaceae.

## MATERIALS AND METHODS

### Sample collection, genome sequencing, and assembly

*Gentiana lawrencei* var. *farreri* was sampled in Qilian Mountain (101°22′33″E, 37°29′53″N, Qinghai, China) from a single plant. Total genomic DNA was isolated from young leaves using a Dzup plant genomic DNA extraction kit (Sangon, Shanghai, China) following the manufacturer's instructions. After DNA isolation, the procedure was performed in accordance with the standard Illumina protocol, including sample preparation and sequencing. Approximately 5–10 μg of genomic DNA was fragmented using ultrasound, which was purified using the CASpure PCR Purification Kit (ChaoShi-Bio, Shanghai, China), followed end repair with poly-A on the 3′ ends. The DNA were then linked to adapters, extracted at specific size after agarose gel electrophoresis and amplified by PCR to yield a sequencing library. Then, a quarter of one flow-cell lane containing the fragmented genomic DNA of *G. lawrencei* var. *farreri* was sequenced using the Illumina HiSeq 4000 platform (Biomarker, Beijing, China), yielding 36.08 million 150-bp paired-end reads from a library of approximately 350-bp DNA fragments. Reads corresponding to plastid DNA were identified using a BLASTN (*E*-value: $10^{-6}$) search against the plastome sequences of two *Gentiana* taxa: *G. straminea* (GenBank accession no. NC_027441) and *G. crassicaulis* (NC_027442). A total of 2,517,802 reads (6.97%) were recovered and assembled using Velvet 1.2.10 (*Zerbino & Birney, 2008*). Eight contigs, ranging in size from 926 to 47,806 bp, were obtained. All the genomic regions located at the junction between the two contigs were verified by Sanger sequencing. The primers used were designed using PRIMER V5.0 and are provided in Table S1. The *G. lawrencei* var. *farreri* plastome sequence was deposited in GenBank (accession no. KX096882).

### Genome annotation

The protein coding genes (PCGs), tRNAs and rRNAs in the cp genome were predicted and annotated using Dual Organellar GenoMe Annotator (DOGMA) using default parameters (*Wyman, Jansen & Boore, 2004*). The  positions of questionable start and stop codons, or intron junctions of the PCGs, were verified using BLAST search against cp genomes of other closely related species. The cp gene map was drawn using OGDraw v1.2 (*Lohse, Drechsel & Bock, 2007*). Simple sequence repeats (SSRs) were detected using MSDB 2.4 (https://code.google.com/archive/p/msdb/downloads) with minimal repeat numbers of 10, 5, 4, 3, 3, and 3 for mono-, di-, tri-, tetra-, penta-, and hexa-nucleotides, respectively.

### Comparative analysis with *G. straminea*

The cp genome sequence from *G. straminea* (NC_027441) was obtained from the National Center for Biotechnology Information (NCBI). Genome comparison to identify the differences between *G. lawrencei* var. *farreri* and *G. straminea* was performed using mVISTA (*Frazer et al., 2004*) and Geneious Basic 5.6.4 (*Kearse et al., 2012*). Nonsynonymous (Ka) to synonymous (Ks) (Ka/Ks) ratios were calculated using DnaSP v5.10 (*Librado & Rozas, 2009*).

## Phylogenetic analysis

To illustrate the phylogenetic relationships of *Gentiana* with other major Gentianales clades with our cp genome sequence, the other 12 available complete cp genomes in the order were downloaded from GenBank (Table S2). *Lactuca sativa* from Asteraceae was used as outgroup. Forty-eight PCGs (*atpA*, *atpB*, *atpE*, *atpH*, *atpI*, *cemA*, *matK*, *ndhD*, *ndhE*, *petA*, *petB*, *petD*, *petG*, *petL*, *petN*, *psaA*, *psaB*, *psaI*, *psaJ*, *psbA*, *psbC*, *psbD*, *psbE*, *psbF*, *psbH*, *psbI*, *psbJ*, *psbK*, *psbL*, *psbM*, *psbN*, *psbT*, *rbcL*, *rpl14*, *rpl16*, *rpl20*, *rpl22*, *rpl33*, *rpl36*, *rpoA*, *rps2*, *rps3*, *rps4*, *rps8*, *rps11*, *rps14*, *rps15* and *rps18*) found in all of the species were extracted from the selected cp genomes. The amino acid sequences of each of the 48 cp PCGs were aligned using MSWAT (http://mswat.ccbb.utexas.edu/) with default settings, and back translated to nucleotide sequences. Phylogenetic analyses were performed using the concatenated nucleotide sequences and PhyML3.1 software (*Guindon & Gascuel, 2003*) using the maximum likelihood (ML) method. PhyML searches relied on the subtree pruning and regrafting (SPR) method with the GTR+I+G model (*p*-inv = 0.404, gamma shape = 0.808), as determined using the Akaike information criterion implemented in jModelTest 2.1.7 (*Guindon & Gascuel, 2003*; *Posada, 2008*). A bootstrap analysis was performed with 100 replications.

# RESULTS

## The overall structure and general features of the *G. lawrencei* var. *farreri* cp genome

The cp genome of *G. lawrencei* var. *farreri* is a closed circular molecule of 138,750 bp (Fig. 1), comprising a pair of IR regions (IRa and IRb) of 24,653 bp, one LSC region of 78,082 bp and one SSC region of 11,365 bp. It has an overall typical quadripartite structure that resembles the majority of land plant cp genomes (*Shinozaki et al., 1986*). The GC contents of the LSC, SSC, and IR regions and the whole cp genome are 35.7, 30.0, 43.6 and 38.0%, respectively, which are similar to the other reported *Gentiana* cp genomes (*Ni et al., 2016*). The cp genome of *G. lawrencei* var. *farreri* contains 130 genes, including 85 PCGs accounting for 66,215 bp, and 37 tRNA and eight rRNA genes accounting for 11,781 bp. Among the 130 genes, 18 are located in the IR region. Most genes are present as a single copy, while all the rRNA genes and some of the tRNA and PCGs in the IR occur as double copies. A total of 84 unigenes were detected in the cp genome and this category is detailed in Table S3. Four genes each have one intron (*atpF*, *rpoC1*, *ndhB* and *rpl2*) and two PCGs (*clpP* and *ndhF*) and 1 ycf (*ycf3*) have two introns. Like most other land plants, *rps12* is trans-spliced, with its two 3′end residues separated by an intron in the IR region, and the 5′end exon is in the LSC region (Fig. 1). The 37 tRNAs contained 30 different tRNA genes and the eight rRNA genes contained four different tRNA genes. Both the number and types of the tRNAs are consistent with those presented in other species of vascular plants (*Shinozaki et al., 1986*).

## Comparison of *G. lawrencei* var. *farreri* and *G. straminea* cp genomes

A comparative analysis between the cp genomes in *Gentiana* revealed that *G. lawrencei* var. *farreri* is 10,241 bp shorter than that of *G. straminea*. As for the four parts of the cp

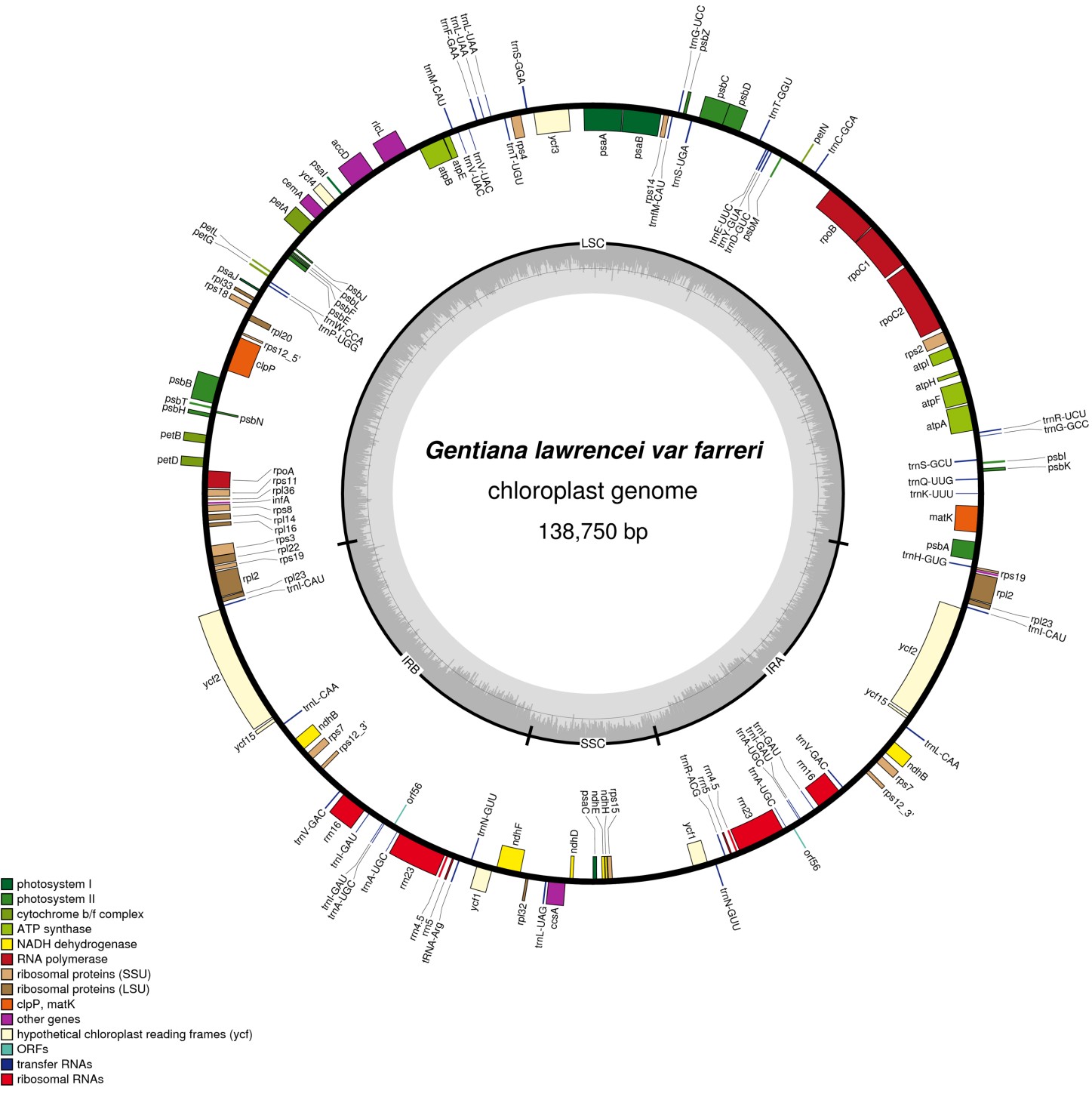

**Figure 1** **Map of the chloroplast genome of *G. lawrencei* var. *farreri*.** Genes drawn inside the circle are transcribed clockwise, and those outside are transcribed counterclockwise. Genes belonging to different functional groups are shown in different colors.
**Table 1  Comparison of genome contents of *G. lawrencei* var. *farreri* and *G. straminea*.**

|  | *G. lawrencei* var. *farreri* | *G. straminea* |
|---|---|---|
| Total sequence length (bp) | 138,750 | 148,991 |
| Large single copy (bp) | 78,082 | 81,240 |
| Inverted repeat region (bp) | 24,653 | 25,333 |
| Small single copy (bp) | 11,365 | 17,085 |
| GC content (%) | 38 | 37.7 |
| Total CDS bases (bp) | 66,215 | 75,780 |
| Average CDS length (bp) | 779 | 758 |
| Total RNA bases (bp) | 11,781 | 11,861 |
| Average intergenic distance (bp) | 467 | 403 |

genome, the LSC, SSC and IR of *G. lawrencei* var. *farreri* are 3185 bp, 5720 bp and 680 bp shorter than those of *G. straminea*, respectively (Table 1). Four big gaps (GapA–D) were detected: GapA (2241 bp) in the LSC, GapB (958 bp) in IRb, GapC (4582 bp) in the SSC and GapD (958 bp) in IRa. The four gaps represent 85% of the "missing" genome. All the gaps were verified by Sanger sequencing with primers designed using PRIMER V5 (Table S1). Compared with *G. straminea*, GapA contains three PCGs (*ndhJ*, *ndhK* and *ndhC*), GapB and GapD contain exon 2 of *ndhB* and GapC contains five PCGs (*ndhG*, *ndhI*, *ndhA* and parts of *ndhE* and *ndhH*). A comparative analysis between *G. lawrencei* var. *farreri* and *G. straminea* cp genomes revealed that the sequence similarities between the *trnH-GUG-psbA*, *trnK-UUU*-italic*trnQ-UUG*, *trnS-GCU-trnG-GCC*, *atpH*-*atpI*, *rpoB-trnC-GCA*, *psbC-trnS-UGA*, *trnT-UGU-trnL-UAA*, *atpB-rbcL*, *ycf1-ndhF*, *rpl32-trnL-UAG* and *trnL-CAA-ycf15* intergenic regions are very low.

## Divergence hotspot

The complete cp genomes of *G. lawrencei* var. *farreri* and *G. straminea* were compared using the mVISTA program to determine the level of sequence divergence. The comparison showed that the coding regions of both cp genomes are highly conserved compared with the noncoding regions. In particular, the intergenic regions showed the greatest divergence between the two cp genomes. More divergence was found in the sequences of *clpP*, *ndhB*, *ndhD*, *ndhE*, *ndhF* and *ndhH*, which are distributed mainly in the SSC regions, compared with other PCGs. The nucleotide and amino acid sequences of the PCGs of *G. lawrencei* var. *farreri* and *G. straminea* are highly similar, with average sequence similarities of 95.0 and 93.0%, respectively. Between the two species, the nucleotide sequence identities of the LSC, SSC, and IR are 88.7, 61.0, and 92.9%, respectively. The most conserved genes include all the rRNA genes, the genes from photosystem I, the cytochrome b/f complex genes and the ATP synthesis genes (Tables S3 and S4).

## Divergence of coding gene sequence

Seventy-four PCGs are shared between the two species. Compared with *G. straminea*, 14 out of the 74 shared PCGs had deletions and six had insertions (Table S4). The average Ks values between the two *Gentiana* species were 0.0551, 0.1133, and 0.0243 in the LSC, SSC, and IR regions, respectively, with a total average Ks of 0.0642 across all regions (Table S4).

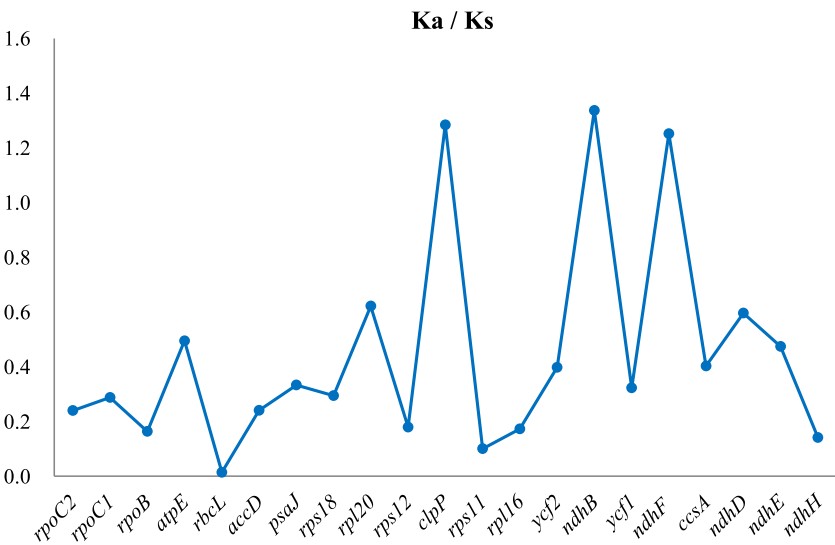

**Figure 2 Gene-specific Ka/Ks ratios between the chloroplast genomes of two *Gentiana* species (*G. lawrencei* var. *farreri* and *G. straminea*).** Three genes (*clpP*, *ndhB* and *ndhF*) returned Ka/Ks ratios greater than 1.0, whereas the Ka/Ks ratios of the other genes were less than 1.0.

Although the coding region is highly conserved, we did observe slight variations. Based on the comparison of Ka/Ks values among the regions, higher Ks values were observed for some genes, including *rps8*, *rpl14*, *rpl36*, *rpl32*, *ndhD*, *rpl36* and *ndhH*. The distribution of Ks values indicated that on average more of genes in the SSC region have experienced higher selection pressures than the rest regions of the cp genome. The Ka/Ks ratio was also calculated, which was >1 for *ndhB* in the IR region, *ndhF* in SSC region and *clpP* from the LSC region (Fig. 2).

## SSR analysis

Thirty-four SSR loci, 394 bp in length, were detected in the *G. lawrencei* var. *farreri* cp genome, and there were 25, three, five, and two mono-, tri-, tetra-, and penta-nucleotide repeats, respectively (Table S5). No dinucleotide repeats were detected in the cp genome. Most of the SSRs are mononucleotide repeats, which is consistent with the study of *George et al. (2015)*. Thirty of the 34 SSRs comprised A and T nucleotides, with a higher AT content (95.9%) in these sequences compared with the rest of the genome. Among the SSRs, 23 were located in intergenic regions and 11 were found in coding genes, including those in the *ccsA*, *rpoC1*, *ndhF*, *atpF*, *rpl32*, *matK*, *rpoA*, *atpB* and *psaB* genes. Compared with *G. straminea*, six loci were identical, five were polymorphic, 28 were lost and 23 were specific to *G. lawrencei* var. *farreri* (Table S5).

## Phylogenetic relationship

An ML phylogenetic tree constructed using 48 PCGs from 12 Gentianales taxa clearly identified the three families (Gentianaceae, Rubiaceae and Apocynaceae) in the analysis as being monophyletic with high bootstrap value (Fig. 3). The tree revealed that *G. crassicaulis* and *G. straminea* are more closely related to one another than either is to *G. lawrencei* var. *farreri*. All the nodes in the tree have high (>95%) bootstrap support.

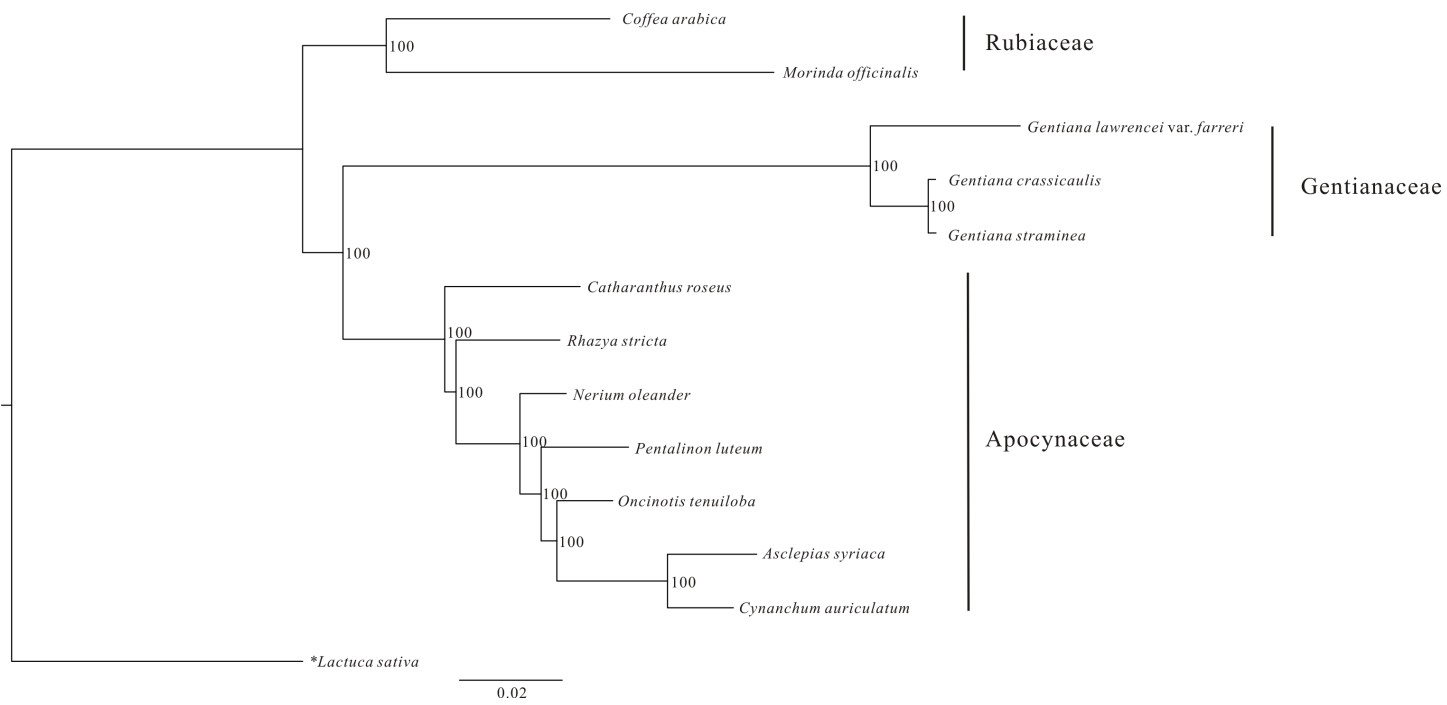

**Figure 3  Phylogenetic analysis of 12 Gentianales species using 48 CDS regions of the chloroplast genomes.** Data sources: *Gentiana straminea* (NC_027441); *Gentiana crassicaulis* (NC_027442); *Catharanthus roseus* (NC_021423); *Rhazya stricta* (NC_024292); *Nerium oleander* (NC_025656); *Pentalinon luteum* (NC_025658); *Oncinotis tenuiloba* (NC_025657); *Cynanchum auriculatum* (NC_029460); *Asclepias syriaca* (NC_022432); *Coffea arabica* (NC_008535); *Morinda officinalis* (NC_028009) and *Lactuca sativa* (NC_007578).

## DISCUSSION

### Evolution of *G. lawrencei* var. *farreri*

Much of the variation in the sequence complexity of angiosperm cp genomes appears to be the result of rather small length mutations. However, our comparative analysis showed that *G. lawrencei* var. *farreri* is 10,241 bp shorter than *G. straminea*. Although the cp genome size is variable, ranging from 120 kb to 160 kb, huge genome losses in congeneric taxa are rarely reported. In general, most of the size changes in angiosperm cp genomes can be accounted for by rare deletions and duplications leading to massive changes in the size of the IR region (*Palmer, 1985*). This is not the case for *G. lawrencei* var. *farreri* and *G. straminea*. The total length variation mainly occurred in the SSC (5720 bp, 55.85%) and LSC (3158 bp, 30.84%) regions rather than the two IR regions (1360 bp, 13.28%). More than half (50.33%) of the sequence length in the SSC region was lost. Therefore, the cp genome size variation in the two *Gentiana* taxa was not caused by deletions in the IR regions, but by deletions in the SSC and LSC regions. Although the IR region can vary from 10 to 76 kb among angiosperms, in the great majority of species it is a rather constant 22–26 kb in size (*Palmer, 1985*). The junction between the IR and LSC region is located within the *rps19* gene in *G. lawrencei* var. *farreri*, similar to majority of dicots and some monocots (*Wang et al., 2008*; *Ni et al., 2016*). The more or less fixed position of IR-LSC junction within a coding gene suggests some selection is operating to constrain the boundaries of

the IR (*Palmer, 1985*). It contributes to the more constant size of the IRs than the LSC and SSC region in the great majority of angiosperms.

The SSC region of *G. lawrencei* var. *farreri* has experienced drastic variation as compared to its congeneric species. Compared with *G. straminea*, the SSC region contributes 55.85% of the cp genome sequence length variation and only showed 61.0% nucleotide identity. The SSC region also has a much higher Ks (0.1133) value than the LSC (0.0551) and IR (0.0243) regions. Two possible explanations about variation in the SSC region were proposed in previous studies. Firstly, the higher rate of molecular evolution in the SSC than other regions was also observed in *Walker, Zanis & Emery (2014)* who attributed it to low proportion of coding *vs*. noncoding regions in the sequence. However, this does not appear to be true in our study. Secondly, the SSC region is a "hotspot" for recombination (*Palmer, 1983*; *Liu et al., 2013*; *Walker et al., 2015*). We did not yet detect inversion in the SSC region of *G. lawrencei* var. *farreri*. Therefore, the drastic variation may be result of other reasons. The functional genes associated with the variation in the SSC region of *G. lawrencei* var. *farreri*, mainly focus on the *ndh* genes, might provide an insight into the reasons for the drastic variation.

In chloroplasts, gene loss is an ongoing process (*Martin et al., 1998*). The huge genome loss in *G. lawrencei* var. *farreri* was mainly accounted for by four big gaps, which caused the loss of the entire *ndhJ*, *ndhK*, *ndhC*, *ndhE*, *ndhG*, *ndhI*, and *ndhA* genes and partial loss of *ndhH* and *ndhB*. The protein products of all the lost genes are NADH dehydrogenase (NDH) subunits. The cp DNA of most of the higher plants contains 11 *ndh* genes, which encode protein subunits of the thylakoid NDH complex. The complex is analogous to mitochondrial complex I (EC 1.6.5.3), which catalyzes the transfer of electrons from NADH to plastoquinone (*Sazanov, Burrows & Nixon, 1998*). The cp *ndh* genes have been retained in most higher plants (*Martín & Sabater, 2010*), but appear to have been lost frequently in parasitic and epiphytic plants (e.g., *Stefanovi & Olmstead, 2005*) along with other cp genes apparently associated with a loss of or reduction in photosynthetic capability (*Iles, Smith & Graham, 2013*). Although the *ndh* genes could be dispensable under mild non-stressing environments, transgenic plants defective in *ndh* genes showed that the NDH complex is required to optimize photophosphorylation rates and showed impaired photosynthesis rates under stress conditions (*Martín & Sabater, 2010*). Cyclic photophosphorylation via the NDH pathway might play an important role in regulating $CO_2$ assimilation under heat-stress conditions, but is less important under chilling-stressed conditions (*Wang et al., 2006*). Therefore, the absence of NDH in *G. lawrencei* var. *farreri* is understandable when considering the cool conditions in the QTP, which is the natural habitat of *Gentiana* (*Ho & Liu, 2001*). Meanwhile, the *ndh* loss between two congeneric species might offer a clue to the divergence and evolution of *Gentiana*.

Variation in the divergence of the coding region was observed between the two *Gentiana* species. Although the coding region was generally highly conserved, the *rps8*, *rpl14*, and *rpl36* genes of the LSC region and the *rpl32*, *ndhD*, *ndhF*, and *ndhH* genes of the SSC region of *G. lawrencei* var. *farreri* showed a higher evolution rate compared with other genes. Based on the sequence identity among the three regions, the IR region is more conserved than the LSC and SSC regions. This agrees with previous studies that hypothesized that the frequent

recombinant events occurring in the IR region result in selective constraints on sequence homogeneity, causing them to diverge at a slower rate than the LSC and SSC regions (*Qian et al., 2013*; *Cho et al., 2015*). Our data confirm a positive selection pressure at the protein coding genes. The *ndhB* gene of the IR region, *ndhF* of the SSC region and *clpP* from the LSC region of *G. lawrencei* var. *farreri* presented higher Ka/Ks ratios (>1.0), indicating that they had evolved under positive selection. The *clpP* gene also showed a high Ka/Ks ratio in *Fagopyrum tataricum* (*Cho et al., 2015*). Interestingly, the *ndhB* and *ndhF* genes experienced positive selection pressure. In the absence of nine *ndh* genes in *G. lawrencei* var. *farreri*, the remaining *ndhB* and *ndhF* genes might play an important role in cyclic photophosphorylation, although the functions of *ndhB* and *ndhF* genes are unknown. The *ndhB* and *ndhF* genes are probably transcribed independently as monocistronic mRNAs (*Martín & Sabater, 2010*). *Favory et al. (2005)* proposed that the transcription of the *ndhF* gene requires the nuclear-encoded sigma4 factor; the *ndhF* product in turn would stimulate the transcription of the other plastid *ndh* genes. Therefore, the selection pressure on the *ndhF* gene may play an important role in evolution of *ndh* genes.

## Phylogenetic value

The ML phylogenetic tree of Gentianales constructed using 48 PCGs clearly grouped the taxa from the three families into three clades. The phylogenetic relationships were consistent with previous studies that classified the three families as three monophyletic clades and identified the Rubiaceae as the base group in the Gentianales (*Backlund, Oxelman & Bremer, 2000*). The cp genome has also been used successfully for phylogenetic reconstruction in several studies (*Carbonell-Caballero et al., 2015*; *Williams et al., 2016*). In *Gentiana*, several phylogeny studies have been carried out (*Yuan & Küpfer, 1997*; *Mishiba et al., 2009*; *Zhang et al., 2009*). However, these studies were all based on one or several DNA fragments, which, together with their complicated evolutionary history, have led to the phylogenetic relationships of *Gentiana* being controversial due to inconsonant sectional classification and the low support for relationships (*Ho & Liu, 2001*; *Favre et al., 2010*). For example, the sect. *Chondrophyllae*, which has 10 series and 163 species, derived within a very short period of time followed by subsequent rapid radiation (*Yuan & Küpfer, 1997*), making the infrasectional phylogenetic relationships of this section difficult to determine. In addition, previous phylogenetic analyses based on internal transcribed spacer regions reclassified five clades in sect. *Cruciata* but failed to find corresponding morphological circumscriptions to support them (*Zhang et al., 2009*). Our analysis also identified substantial length variation and amount of base substitutions in the cp genome between two species of *Gentiana*; therefore, to realize the full potential of the cp genome in phylogenetic analysis, more taxa of different secttions should be included in the cp genome comparison analysis.

Chloroplast SSRs are good tools for studies in plant ecology and evolution (*Provan, Powell & Hollingsworth, 2001*). Microsatellites often show high levels of polymorphism and are thus used widely in studies of genetics and evolution. However, SSRs in the nuclear genome are usually species-specific and are thus used mainly for intraspecific genetic studies rather than phylogenetic studies of related species. Unlike nuclear SSRs, chloroplast SSRs are frequently cross-amplified in related species and thus could be used for

phylogenetic studies (*Provan, Powell & Hollingsworth, 2001*). We detected five polymorphic SSRs between *G. lawrencei* var. *farreri* and *G. straminea*, which belong to different sections. SSRs are more polymorphic than cp loci that are amplified by universal primers; therefore, the polymorphic SSRs could offer higher resolution for phylogenetic tree construction in *Gentiana*.

## CONCLUSION

We present the first report of the complete cp genome sequence of *G. lawrencei* var. *farreri* and describe its evolutionary characteristics in comparison with *G. straminea*. About 10kb sequence which mainly consistent of nine *ndh* genes were lost in *G. lawrencei* var. *farreri*. The divergence hotspots and SSRs clarified here could be used as molecular markers and will be useful for further studies on population genetics, phylogenetics and evolution of the genus *Gentiana*.

## ACKNOWLEDGEMENTS

We thank Shan-shan Sun of the Wuhan Botanical Garden, Chinese Academy of Sciences, for providing laboratory support.

### Funding

This work was supported by the National Natural Science Foundation of China (Grant No. 31600296), Science and Technology Project of Henan Province, China (No. 162102110097) and Educational Commission of Henan Province (No. 16A210033). The funders had no role in study design, data collection and analysis, decision to publish, or preparation of the manuscript.

### Grant Disclosures

The following grant information was disclosed by the authors:
National Natural Science Foundation of China: 31600296.
Science and Technology Project of Henan Province, China: 162102110097.
Educational Commission of Henan Province: 16A210033.

### Competing Interests

The authors declare there are no competing interests.

### Author Contributions

- Peng-Cheng Fu conceived and designed the experiments, performed the experiments, contributed reagents/materials/analysis tools, wrote the paper.
- Yan-Zhao Zhang performed the experiments, contributed reagents/materials/analysis tools, prepared figures and/or tables.
- Hui-Min Geng performed the experiments, analyzed the data, prepared figures and/or tables.
- Shi-Long Chen analyzed the data, reviewed drafts of the paper.

## DNA Deposition

The following information was supplied regarding the deposition of DNA sequences:
  GenBank: KX096882.

## Data Availability

  The raw data has been supplied as a Supplementary File.

## Supplemental Information

Supplemental information for this article can be found online at http://dx.doi.org/10.7717/peerj.2540#supplemental-information.

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
