# Peer review of "The complete chloroplast genome sequence of Gentiana lawrencei var. farreri (Gentianaceae) and comparative analysis with its congeneric species"

_PeerJ, doi:10.7717/peerj.2540_

## Round 0.1 · original submission · Major Revisions

· Academic Editor

Major Revisions

Your manuscript “The complete chloroplast genome sequence of Gentiana lawrencei var. farreri (Gentianaceae) and a comparative analysis with its congeneric species” has been seen by two qualified reviewers. Based on their recommendation and my own detailed assessment I feel this manuscript is potentially suitable for publication in PeerJ but requires after extensive revisions and reanalysis. I have summarized the major issues that must be addressed below before reconsideration, but the authors should provide detailed responses to all of the reviewer queries.
1. There are many essential genes missing from the Gentiana chloroplast genome including several genes within the IR which are found in one IR region but not the other. The IRs should be nearly identical suggesting either an assembly or annotation issue. The authors should validate the chloroplast assembly and redo the annotation (including manual curation) to ensure that the missing genes are in fact missing.
2. The methods are vague and it would be difficult to replicate this study using the detailed provided here.
3. There a number of grammatical errors throughout the text that need to be corrected.

·

Basic reporting

Overall Language

The overall format seemed to follow the guidelines, however, there are numerous grammatical mistakes and it would be useful for this to be thoroughly read over.

Intro and Background

Lines 19-21 do not seem correct, in line 180 of the paper the author states that the IR can be up to 76kb which if line 20 is true then the SSC and LSC would need to be extremely small. I would suggest reviewing those references.

Line 189-190 the Walker, Zanis and Emery Citation is incorrect, that was not a suggestion from the study. It would be more applicable to have the citation after line 188 as it showed that the SSC, LSC and IR differ in rates of molecular evolution, similar to how the authors found difference in Ks values for the SSC, LSC and IR.

Line 192 miss cites Walker et. al 2015. In Walker et. al 2015 they argue against the SSC being a hotspot for inversion events on an evolutionary scale, as the SSC is found in both directions in the plant cells. For further information on this a good resource is: Palmer, 1983 "Chloroplast DNA exists in two orientations"

Line 196 it is difficult to tell what the authors mean by gene loss is an on going process, are they talking about in chloroplasts? There should be a citation after this or a more thorough explanation.

Line 225 to 227 there should be some form of correction for multiple comparisons such as a bonferroni correction, finding a Ka/Ks ratio above 1 could easily be a by product of having performed the test on so many samples.

Line 244 should specify what is meant by rapid evolution. Is this a rapid molecular rate or some form of radiation.


Figures

For Figure 1. It’s bothering me that so many of the chloroplast genes are missing. Your IR regions should be exactly the same just reverse complement, yet IRA is missing the YCF2 gene and IRB is missing ndhB and rps7. I would strongly recommend going through and making sure this is all annotated correctly.

For Figure 2. I’m not entirely sure why you are showing the Ks values here, it may help the figure if you separated the genes into the IR, SSC and LSC regions and showed the difference in Ks values for those. You would likely see a dip in the Ks values for those from the IR. Also, if you are doing a multiple comparison such as this it would be good to include some sort of correction for multiple comparisons, such as a bonferroni correction, since when you do this many comparisons by chance alone you would expect some of these to have a Ka/Ks value above 1.

Figure 3. This tree looks to be unrooted, you have your outgroup forming a polytomy with multiple clades, the tree should be properly rooted on the outgroup before you make any inference regarding the relationships.

Experimental design

Overall as the experimental design is written out now, I would not be able to replicate the study. It would be very useful for the authors to give a more thorough explanation of the programs they used.

There are a few times that programs are mentioned in the results but are not put in the methods, such as mVISTA is mentioned on line 131 but not mentioned anywhere else.

On line 134 ndhE is said to have a high amount of divergence but it is said to be missing as part of GapC on line 124.

For the Genome annotation: I don’t think you can verify start and stop codons using BLAST, BLAST is typically left to be just a homology search tool. Do the authors mean that they searched Genbank and compared their start and stop codons to closely related species?

For the Comparative analysis: I am not sure how the authors were able to analyze nucleotide and amino acid diversity from BLAST. If they could elaborate on that, it would be very helpful. Also, I believe Geneious is a set of programs, if the authors could elaborate on what setting or which programs in Geneious they used to identify differences that would be very helpful.

For the Phylogenetic analysis: It seems like in creating the phylogeny the authors concatenated all the genes together and built the phylogeny out of that. However, we know that the different regions have different rates of evolution and therefore it would be highly valuable for the authors to re-run the phylogenetic analysis with the data partitioned by gene. This will allow a separate model of evolution to be inferred for each gene and help lower some of the bias created by using genes that evolve at different rates. The authors also state they use the GTR model of evolution, which should be great, but they should also account for invariability of sites using either INV or Gamma, both options should be available in PhyML.

In continuing with the phylogenetic analysis although the authors used an outgroup, based on Figure 3 they appear to have never rooted the tree.

Validity of the findings

A lot of the analyses appear to need to be re-examined, such as the un-rooted phylogenetic tree (Fig. 3), the missing genes in the chloroplast IR (Fig 1.) and the use of BLAST as a means of assessing nucleotide and amino acid diversity, instead of leaving it as a tool for homology identification.

Regarding the conclusions, the authors appear to have a very interesting story regarding gene loss in the chloroplast; however, this is not emphasized in the conclusion. While the paper has some extensive discussion regarding gene loss, the conclusion fails to mention this.

Additional comments

You have a very interesting story regarding gene loss in the chloroplast and appear to have created a valuable genomic resource, however, there appears to be a number of issues with the analyses that you ran. I would strongly suggest re-running many of the analyses and doing a more thorough check on the ndh gene loss. Best of luck with your study!

Reviewer 2 ·

Basic reporting

My main concern is that the study does not appear to be a logical unit for publication. High-throughput sequencing and increasing computational sophistication in the bioinformatics community have made the assembly of complete chloroplast genomes a highly automated process. That is not to say a single chloroplast genome cannot be worthy of a full analysis, but I do not feel a strong case was made for such an exception here. Several of the results in this analysis identify the chloroplast genome of Gentiana lawrencei var. farreri as possessing a structure consistent with that of most land plants. These observations are unnecessary and dilute the more substantial results of the study. The article could potentially be improved by providing more background on particular features of the cp genome as they relate to Gentiana and then focusing on these features in the analyses rather than listing characteristics of the genome that are not obviously connected to any particular research question. Given the content currently included in the article, it would more appropriately be presented as a brief 2-page note.

Experimental design

Substantial improvements could be made in outlining the research question(s) addressed by sequencing the chloroplast genome of Gentiana lawrencei var. farreri. Many potentially interesting and meaningful features of the genome are identified in the article, but the conceptual framework underlying them is frequently missing. It is also unclear why certain analyses were conducted while others were not. Why was G. crassicaulis not included in comparative analyses? The comparative section of the study has the potential to be expanded into an appropriate publication unit, especially if other species in Gentianaceae were included. The purpose of the phylogenetic analysis is also unclear. As far as I can tell, the only novel finding from the tree is that G. crassicaulis and G. straminea are more closely related to one another than either is to G. lawrencei var. farreri, but this result is not highlighted by the authors and was presumably identified in previous phylogenies.

Validity of the findings

The results as presented appear to me to be sound, although some additional details on methods would be helpful. The main issue with conclusions are that they are sometimes vague, suggesting the chloroplast genome presented here will help conservation efforts in the genus Gentiana. Such a conclusion may be true, but it is not at all obvious why the chloroplast genome would be particularly useful for that purpose. Again, there is a disconnect between reported results and the focus of the study, so the article lacks logical cohesion. Limiting the features of the chloroplast genome that are highlighted and supporting them with an expanded biological context would make the study a potentially valuable contribution beyond the production of a single plastome.

Additional comments

I did not see anything wrong with the data you generated, and I found some of the comparative analyses interesting. However, given the relative ease with which chloroplast genomes can be sequenced, assembled, and annotated, publication of a single genome needs to be supported by a strong conceptual framework and guided by a clear set of research questions. I think it is possible to do this with your study by including the other species of Gentiana (and possibly other Gentianaceae) in your comparative analyses and focusing the introduction and discussion on a subset of features.

Annotated reviews are not available for download in order to protect the identity of reviewers who chose to remain anonymous.

---

## Round 0.2 · Minor Revisions

· Academic Editor

Minor Revisions

Your revised manuscript has undergone an additional round of review. Based on the reviewers assessment and my own, there are still some revisions that need to be made before this manuscript is suitable for publication in PeerJ.

·

Basic reporting

Since the first submission the article has improved greatly, however, their are still some areas of concern.

Figures: Figure 3 appears to still be unrooted, this tree should be rooted on Lactuca, as of now the outgroup forms a polytomy with the other clades.

Line 30: Should be “genome has been”

Line 34: What is being implied by several DNA fragments? This could indicate similar issues to your study as a chloroplast is basically several DNA fragments.

Introduction should mention something regarding the ndh gene loss

In line 143 what is meant by most conserved genes, looking at Table S3 only genes are listed, was this meant to be Table S4?

I’m having trouble with Table S5, do you mean identical instead of identity? Also, it would be good to clear up the + and the -.

It would be good to have a line or something highlighting the monophyletic clades in figure 3.

Have you used PCR to try and verify the potential areas where the ndh genes appear to be missing? Maybe design primers that would be on either side of the missing genes and see if they amplify a smaller fragment or if they do amplify the missing gene, it seems odd that it is missing from your species and not the closely related species.

In line 203 it would be more accurate to refer to this as “hotspot for recombination” rather than a “hotspot for inversion events”.

In line 249 I don’t think you can make this statement as the tree is still unrooted

Experimental design

The authors have significantly improved on their explanation, however, the phylogenetic analysis still appears to use an unrooted tree

Validity of the findings

No comment

Additional comments

Overall, the study appears to be a lot stronger than the first draft. The story of the ndh genes is very interesting but more validation would be helpful to make statements regarding gene loss. Best of luck with your study!

---

## Round 0.3 · accepted · Accept

· Academic Editor

Accept

The revisions have addressed the previous concerns and the manuscript is now suitable for publication in PeerJ.